# Excellent Photoelectro-Catalytic Performance of In$_2$S$_3$/NiFe-LDH Prepared by a Two-Step Method

Xiaona Liu [1,2,3,*,†], Zhenzhen Li [1,†], Wenxia Liu [1,*], Huili Wang [1], Zhaoping Song [1,3], Dehai Yu [1] and Guodong Li [1]

[1] Key Laboratory of Pulp and Paper Science & Technology of Ministry of Education, State Key Laboratory of Biobased Material and Green Papermaking, Faculty of Light Industry, Qilu University of Technology, Shandong Academy of Sciences, Jinan 250353, China; 18364196285@163.com (Z.L.); zsong@qlu.edu.cn (Z.S.); yudehai@qlu.edu.cn (D.Y.); lgd@qlu.edu.cn (G.L.)

[2] Shandong Environmental Protection Development Group Co., Ltd., Jinan 250101, China

[3] National Forestry and Grassland Administration Key Laboratory of Plant Fiber Functional Materials, Fuzhou 350108, China

[*] Correspondence: lxn511944@qlu.edu.cn (X.L.); liuwenxia@qlu.edu.cn (W.L.)

[†] These authors contributed equally to this work.

**Abstract:** In this work, we synthesize hierarchical In$_2$S$_3$/NiFe-layered double hydroxide (In$_2$S$_3$/NiFe-LDH) nanoarrays on an F-doped SnO$_2$ glass substrate via a two-step method, which the In$_2$S$_3$ electrode film was firstly prepared using chemical bath deposition on F-doped SnO$_2$ glass substrate, and then the layered NiFe-LDH was deposited on In$_2$S$_3$ electrode film by hydrothermal synthesis. The two-component photoanode In$_2$S$_3$/NiFe-LDH exhibits significantly enhanced photoelectrochemical properties compared with the In$_2$S$_3$ single-component; due to that, the NiFe-LDH nanosheets depositing on the surface of In$_2$S$_3$ nanocrystal can reduce the accumulation of photogenic holes, facilitate the separation of photogenerated charge carriers, and enhance the light response and absorption. After being decorated with the NiFe-LDH nanosheets, the In$_2$S$_3$/NiFe-LDH photoanode displays a lower onset potential of 0.06 V and an enhanced photocurrent density as high as 0.30 mA·cm$^{-2}$ at the potential of 1.0 V (vs. RHE). Furthermore, it also displays a 90% degradation rate of xylose oxidizing into xylose acid in 3 h under UV light. This work provides a promising approach for designing new heterojunctions applied to biomass degradation.

**Keywords:** photoelectrocatalysis; In$_2$S$_3$/NiFe-LDH; hierarchical; oxidation of xylose; biomass degradation

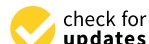



## 1. Introduction

The direct conversion of solar energy into chemical energy using semiconductor photoelectrodes has drawn a lot of attention for many decades, particularly in the removal of organic pollutants in water via photocatalysis technology [1–7]. Photocatalysis was defined as a process in which the photocatalyst absorbs light energy and creates the photogenerated electrons and holes that participate in redox reactions, thus achieving the purpose of purifying pollutants, synthesis, and transformation of substances [8–10]. Photoelectrocatalytic methods, combining electrochemical and photocatalysis technologies, have recently raised great interest due to their high efficiency and low cost in removing persistent pollutants in wastewater [11,12]. The photoelectrocatalysis under the action of an electric field inside the semiconductor can further promote the separation of photogenerated carriers, thereby improving the catalytic oxidative degradation efficiency [11,12]. The photocatalytic reaction is a pair of simple conjugated redox reactions in which the role of oxygen is very important because oxygen acts as a receptor for photogenerated electrons and can restrain the recombination of photo-induced carriers [11,12]. The photoelectrocatalytic method of applying an electric field can cause the oxidation reaction and the reduction reaction to occur in different places. The reduction reaction can be either an oxygen reduction in a

solution or a hydrogen evolution reaction, which is different from the usual photocatalytic system. However, almost all photoelectrocatalytic degradation studies have been limited to the oxidation of photogenerated holes [13,14] and often neglected the photoreduction electrons during oxidative degradation. Up to now, several photocatalysts with outstanding properties have been found [15], such as titanium dioxide ($TiO_2$) [16] and ZnO [17] in the UV spectral range and $WO_3$ [18], CdS [19], graphene-based hybrid semiconductors, and $In_2S_3$ in the visible range.

Among various photocatalysts reported, indium sulfide ($In_2S_3$), due to its low toxicity, high carrier mobility, and stability at ambient conditions, has attracted special interest and been considered a promising candidate for optoelectronic and photovoltaic applications [20]. Although $In_2S_3$ is an ideal photoanode light absorber, the charge separation efficiency still needs to be improved. In order to efficiently utilize solar energy, improve the separation of photo-induced carriers, and promote the redox reactions, there are many approaches, including engineering the electronic structure by doping metal or nonmetal elements [21,22], designing nanostructure of the photoanodic surfaces [23], and constructing heterojunction composite by coupling two energy level-matching semiconductors [24–26]. $IrO_2$ and $RuO_2$ were once employed as catalysts to improve photocatalytic activity, but this strategy is limited due to its high cost [27]. Layered double hydroxide (LDH) material is a kind of anionic clay, with the layered framework consisting of the divalent and trivalent metal cations uniformly distributing and the anions intercalating [28,29]. Therefore, transition metal (oxy)hydroxides, such as $Ni(OH)_2$ [30], CoFe layered double hydroxide (CoFe-LDH), and NiFe-layered double hydroxide (NiFe-LDH) [31–34], have been widely applied due to their evident enhancement in the water oxidation efficiency by alleviating photogenerated holes accumulation and enhancing oxygen evolution kinetics.

Xylose, an abundant pentose sugar present in lignocellulosic plant material, is currently considered a potential primary source for ethanol or xylitol production used in many fields, such as the synthesis of various biodegradable chemicals, chelating agents, additives for cement, concrete, food, and pharmaceutical, etc. [35–39]. Therefore, scientists have developed many different chemical and biological methods attempting to oxidize xylose into xylose acid and other chemicals. Mervi Toivari and his team reported the metabolic engineering of saccharomyces cerevisiae for the bioconversion of D-xylose to D-xylonate, while others achieved xylonic acid from xylose via using bacteria [40]. Chen et al. designed a $TiO_2/Ti_3C_2$(TT) photocatalyst for the oxidation of xylose to D-xylonic acid under ambient conditions, exhibiting 64.2% of D-xylonic acid yield with excellent stability [41]. Li et al. reported that $ZnS@Bi_2S_3$ nanosheets prepared by one-step controllable, which were used for the highly selective oxidation of biomass-derived monosaccharides to produce value-added sugar acids [42]. The microbial conversion of xylose has good selectivity; however, the conversion efficiency is still low, and it is difficult to separate the microbes and control the generation of by-products, so it is of great significance to seek new approaches to settle these problems. Now, there are two chemical methods for the oxidation of xylose into xylonic acid. One is that xylose is oxidized in aqueous media with the catalysts of precious metal Pd or Pt supported on Mg, $\gamma$-$Al_2O_3$, carbon black, etc., in the presence of oxygen, air, and hydrogen peroxide [43]. Mirescu and Prüe reported that the selectivity of 4.6% $Pd/Al_2O_3$ catalyst reached 99% for the oxidation of xylose and that 0.45% $Au/TiO_2$ catalyst was higher than 99.5% under the same conditions [44]. The other method is electrochemical oxidation, which does not require the addition of additional oxidants and can be activated on solids. The reaction activity can be controlled precisely by changing the structure/composition of the catalyst, and the reaction selectivity can be improved by adjusting the anode potential (battery voltage) to control the adsorption/activation of the water and sugar on the surface of the catalyst [43,45–47]. However, few studies have been conducted on this presently, and only platinum and gold have been investigated extensively, ascribing to their good activity in electrochemical reactions.

Based on the above, we propose an electrochemical oxidation-like method, a photo-electrochemical method, applying for the oxidation of xylose into xylose acid in this work.

The first step is to choose the appropriate photocatalytic material. Nanostructured semiconductor materials prepared on fluorine-doped tin oxide (FTO) or fluorine-doped indium tin oxide (FITO) substrates possess higher adhesion and electron transfer efficiency [48]. Moreover, layered double hydroxide (LDH) directly hydrothermally grown on semiconductor photocatalysts also exhibits better adsorption and higher activity. In this work, the hierarchical $In_2S_3$/NiFe-layered double hydroxide ($In_2S_3$/NiFe-LDH) nanoarrays on an F-doped $SnO_2$ glass substrate via a two-step method was synthesized, whereby the $In_2S_3$ electrode film was firstly prepared using chemical bath deposition on F-doped $SnO_2$ glass substrate, and then the layered NiFe-LDH was deposited on $In_2S_3$ electrode film by hydrothermal synthesis. The two-component photoanode $In_2S_3$/NiFe-LDH exhibits significantly enhanced photoelectrochemical properties compared with the $In_2S_3$ single-component; due to that, the NiFe-LDH nanosheets depositing on the surface of $In_2S_3$ nanocrystal can reduce the accumulation of photogenic holes, facilitate the separation of photogenerated charge carriers, and enhance the light response and absorption. After being decorated with the NiFe-LDH nanosheets, the $In_2S_3$/NiFe-LDH photoanode displays a lower onset potential of 0.06 V and an enhanced photocurrent density as high as 0.30 mA·cm$^{-2}$ at the potential of 1.0 V (vs. RHE). Furthermore, the prepared $In_2S_3$/NiFe-LDH semiconductor material was used for photoelectrodes to degrade xylose, and it displayed excellent degradation performance under both ultraviolet light and visible light, with an oxidation rate higher than 90% within 3 h under UV light, indicating that the prepared $In_2S_3$/NiFe-LDH semiconductor material can successfully oxidize xylose into xylose acid. This work provides a promising approach for designing new heterojunctions to apply to biomass degradation.

## 2. Results and Discussion

### 2.1. XRD Structural Characterization

The XRD patterns of the pristine $In_2S_3$ and $In_2S_3$/NiFe-LDH are shown in Figure 1. All the diffraction peaks in Figure 1a are in good consistency with the standard data of the cubic structure $In_2S_3$ (JCPDS no.32-0456) [49], which is in accordance with SEM, and the sharp diffraction peaks imply that the $In_2S_3$ sample was highly crystallized. As is shown in Figure 1b, there exist the diffraction peaks not only of cubic structure $In_2S_3$, denoted by dots, but also of NiFe-LDH (JCPDS no.38-0715) [29], denoted by rhombuses, which indicates that the composite of $In_2S_3$ and NiFe-LDH was successfully synthesized using the two-step method. No characteristic peaks of any impurities were detected.

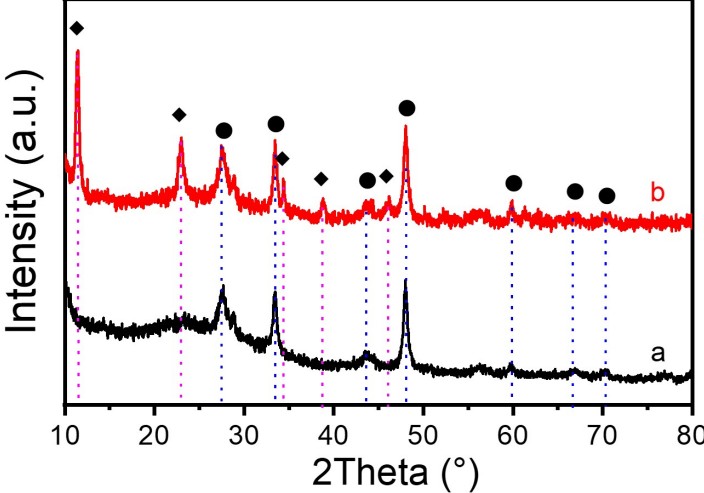

**Figure 1.** XRD patterns of (a) $In_2S_3$ and (b) $In_2S_3$/NiFe-LDH. Insets: Peaks due to $In_2S_3$ (●) and NiFe-LDH (◆).

### 2.2. SEM and TEM Morphology Characterization

The morphologies of pristine $In_2S_3$, NiFe-LDH, and $In_2S_3$/NiFe-LDH were observed using SEM and TEM. In Figure 2a, the cubes of $In_2S_3$ are clearly observed, which further proves the cubic phase of $In_2S_3$. Figure 2b,c shows the morphology of the synthesized Ni-Fe hydrotalcite and $In_2S_3$/NiFe-LDH, respectively. The sample was rapidly dried in an oven at a high temperature, resulting in slight damage and cracking of the NiFe-LDH on the surface layer, and thus the double-layered material deposited on the FTO can be observed in Figure 2c. It can clearly be seen that the grain size of $In_2S_3$ is obviously larger than that of NiFe-LDH in Figure 2a,b, and there are thin NiFe-LDH films grown uniformly on the $In_2S_3$ displayed in Figure 2e. At the same time, in order to observe the growth thickness of $In_2S_3$, we photographed the cross-section of the material. It can be seen from Figure 2d that the thickness is about 1 μm. The synthesized $In_2S_3$/NiFe-LDH is further characterized by TEM, EDX, and mapping. Figure 2e exhibits the cubic shape of synthetic $In_2S_3$ and the layered structure of NiFe-LDH. The HRTEM images in Figure 2f of the $In_2S_3$/NiFe-LDH reveal that the fringe spacing of 0.319 nm and 0.257 nm agree well with the spacing of the lattice plane (311) of $In_2S_3$ and lattice plane (012) of NiFe-LDH, respectively, indicating the formation of NiFe-LDH on the surface of $In_2S_3$. The EDS elemental analysis in Figure 2g displays the presence of In, S, Ni, Fe, and O elements and uniform signals of In, S, Ni, Fe, and O elements in the SEM-EDS elemental mapping images (as shown in Figure 2h–l) demonstrates the homogeneous distribution of NiFe-LDH on the $In_2S_3$ surface. These results suggest that $In_2S_3$ can be well decorated with the NiFe-LDH, which is favorable for facilitating the charge transfer between the $In_2S_3$ and NiFe-LDH.

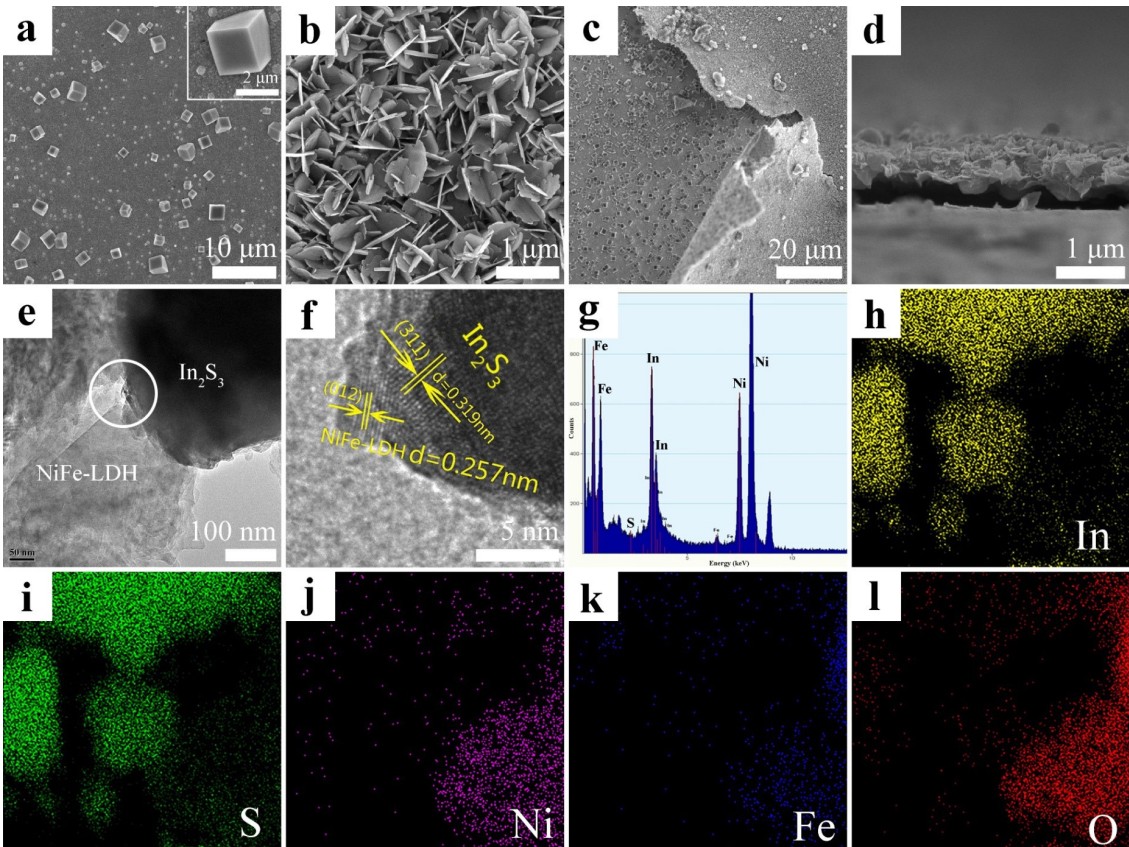

**Figure 2.** SEM images of the $In_2S_3$ (**a**), NiFe-LDH (**b**), and $In_2S_3$/NiFe-LDH (**c,d**). HRTEM images of $In_2S_3$/NiFe-LDH (**e,f**). Energy dispersive spectroscopy (EDS) spectra (**g**) and mapping (**h–l**) of $In_2S_3$/NiFe-LDH.

### 2.3. XPS Characterization

The surface chemical composition and valence states of the as-prepared $In_2S_3$/NiFe-LDH were investigated using X-ray photoelectron spectrometry (XPS) measurements, as shown in Figure 3, and the binding energies obtained in the XPS analysis were corrected for specimen charging by reference to a C 1s value of 284.6 eV. Figure 3a provides the general survey spectrum of $In_2S_3$/NiFe-LDH, which demonstrates the existence of In, S, Ni, Fe, O, and C elements in the sample. The observed C peak at 284.6 eV may be ascribed to an adventitious carbon-based contaminant, which is used to calibrate all of the binding energies, while the O element detected in XPS spectra could be oxygen from Ni- and Fe-hydroxides. The XPS spectrum of the $In_2S_3$ sample is similar and consistent with the typical $In_2S_3$ spectrum reported in the literature [50]. The XPS spectrum of the In 3d region is displayed in Figure 3b, consisting of two characteristic peaks with binding energies of 445.2 eV and 452.8 eV, which correspond to the signals from doublets of In $3d_{5/2}$ and In $3d_{3/2}$ in the trivalent oxidation state, respectively. The peaks, located at 161.8 and 163.0 eV, as shown in Figure 3c, could be assigned to the $-2$ valence state of sulfur for the S $2p_{3/2}$ and S $2p_{1/2}$, respectively [51]. Figure 3d shows the high-resolution XPS peaks of Ni element, exhibiting the Ni 2p spectrum with two peaks at the binding energies of 855.7 and 873.4 eV, which can be assigned to $2p_{3/2}$ and $2p_{1/2}$ peaks in NiFe-LDH, respectively. The Fe 2p spectrum in Figure 3e shows two peaks at 704.1 and 724.2 eV, which are assigned to Fe $2p_{3/2}$ and Fe $2p_{1/2}$ in NiFe-LDH, respectively [52]. As displayed in Figure 3f, the characteristic peak of O1s around 530.5 eV comes from the overlapping contributions of oxide ions Gaussian fitted into three peaks: at 530.3, 531.0, and 531.7 eV, which were contributed to $O^{2-}$ species, chemisorbed activity ($O_2^{2-}$ and $O^-$), and adsorbed molecular water, respectively [53,54]. The XPS results prove that the formation of NiFe-LDH in the $In_2S_3$/NiFe-LDH composite occurs, which is in good accordance with the results of the XRD, SEM, and HRTEM analyses.

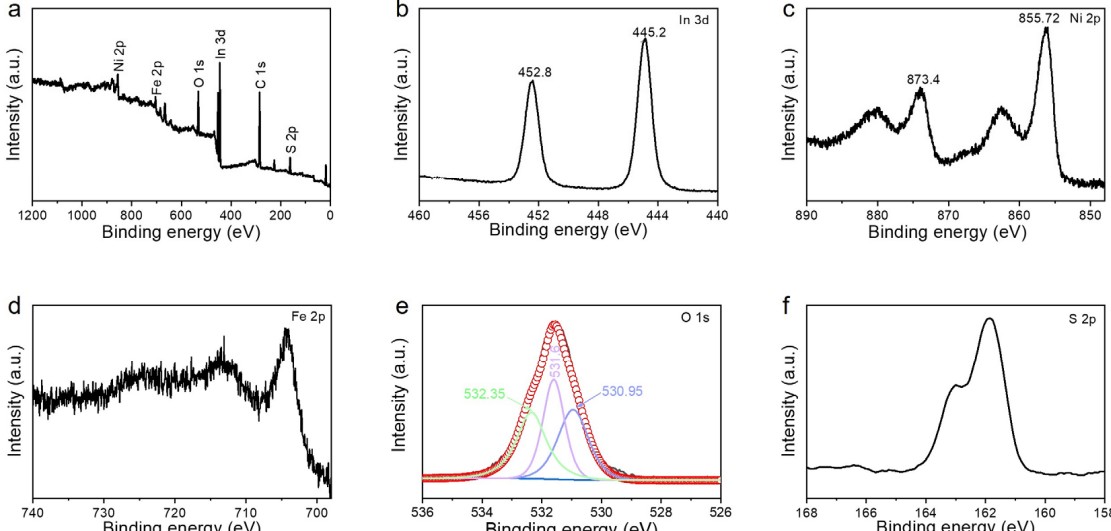

**Figure 3.** (**a**) XPS–fully scanned spectra, (**b**) In 3d spectra, (**c**) S 2p spectra, (**d**) Ni 2p spectra, (**e**) Fe 2p spectra, and (**f**) O 1s XPS spectra.

### 2.4. UV–Vis Absorption Spectra

The UV–vis absorption spectra in Figure 4 were used to investigate the optical properties of $In_2S_3$, NiFe, and $In_2S_3$/NiFe-LDH. All the samples were scraped from the FTO conductive glass for the convenience of testing. It is obvious that the $In_2S_3$ sample has significant absorption in a wavelength of less than 630 nm and weak absorption in a wavelength greater than 630 nm. After being decorated with NiFe-LDH nanosheets, the visible light absorption of $In_2S_3$/NiFe-LDH increases compared with that of NiFe-LDH, demon-

strating that it has good visible light response ability. The band gap can be calculated by $Eg = 1240/\lambda$, where the $\lambda$ is the optical absorption edge of the semiconductor. The optical absorption edges of $In_2S_3$ and NiFe are 619 and 780 nm, and their corresponding band gaps are calculated to be 2.00 and 1.59, respectively, in line with the previous reports [55,56].

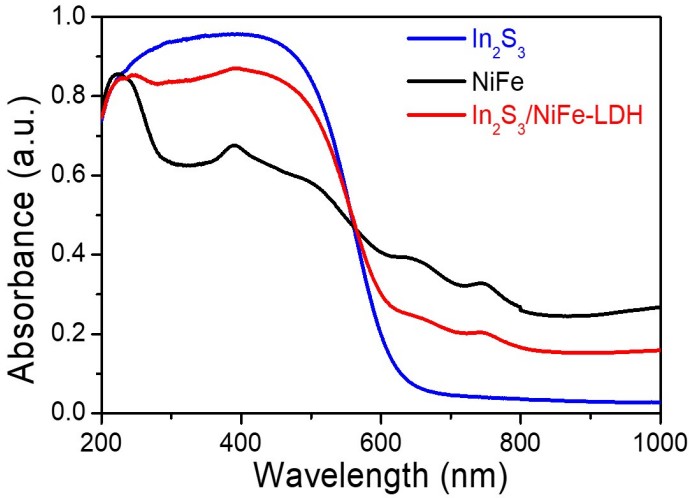

**Figure 4.** UV–Vis diffuse reflectance spectra of the samples.

## 2.5. Photoelectrochemical (PEC) Performance

To study the PEC performance of NiFe-LDH nanosheet arrays loaded on the $In_2S_3$ photoanode, the samples were fabricated as photoanodes with a well-defined area of 1 cm$^2$. All PEC properties were measured under AM 1.5 G simulated sunlight (100 mW cm$^{-2}$) in a standard three-electrode electrochemical cell with a Pt foil as the counter electrode, an Ag/AgCl as the reference electrode, and a 0.25 M $NaSO_4$ solution as the electrolyte. According to previous reports [57,58], the tests were measured under the backlight to obtain a better photocurrent, ascribing to the photogenerated electrons having to travel longer distances collected by FTO back contact under front-side illumination. The relationship between current density and voltage of prepared $In_2S_3$/NiFe-LDH optoelectronic materials was studied by linear scanning voltammetry (LSV). As shown in Figure 5a, the current density of the simple FTO conductive glass sheet is almost zero, and that of the single-layer of $In_2S_3$ photoelectric material, whose maximum current density is just 0.1 mA/cm$^2$ in the voltage range of 0–1.0 V, is also relatively small. At the same time, the current density of $In_2S_3$/NiFe-LDH optoelectronic material increases obviously, about three times higher than that of $In_2S_3$ at 1.0 V voltage, after the NiFe-LDH nanosheet arrays grown on the surface of $In_2S_3$, which means that more photo-induced electrons are transferred to the opposite electrode via the external circuit $In_2S_3$/NiFe-LDH nanomaterials. The Mott–Schottky test can determine the type of semiconductor, the carrier concentration, and the flat band potential. Therefore, in order to evaluate the conductor types of the synthesized $In_2S_3$/NiFe-LDH optoelectronic materials and to further study the charge separation process on the electronic properties, the Mott–Schottky curves of $In_2S_3$/NiFe-LDH optoelectronic materials were tested under dark conditions. Three frequencies of the $In_2S_3$/NiFe-LDH optoelectronic materials were tested using the Mott–Schottky test to explore the influence of different frequencies on the materials. As shown in Figure 5b, all curves show positive slopes, indicating that the $In_2S_3$/NiFe-LDH are n-type semiconductors, and the space charge layer differential capacitance of the semiconductor is different at different frequencies under the same electric potential [59].

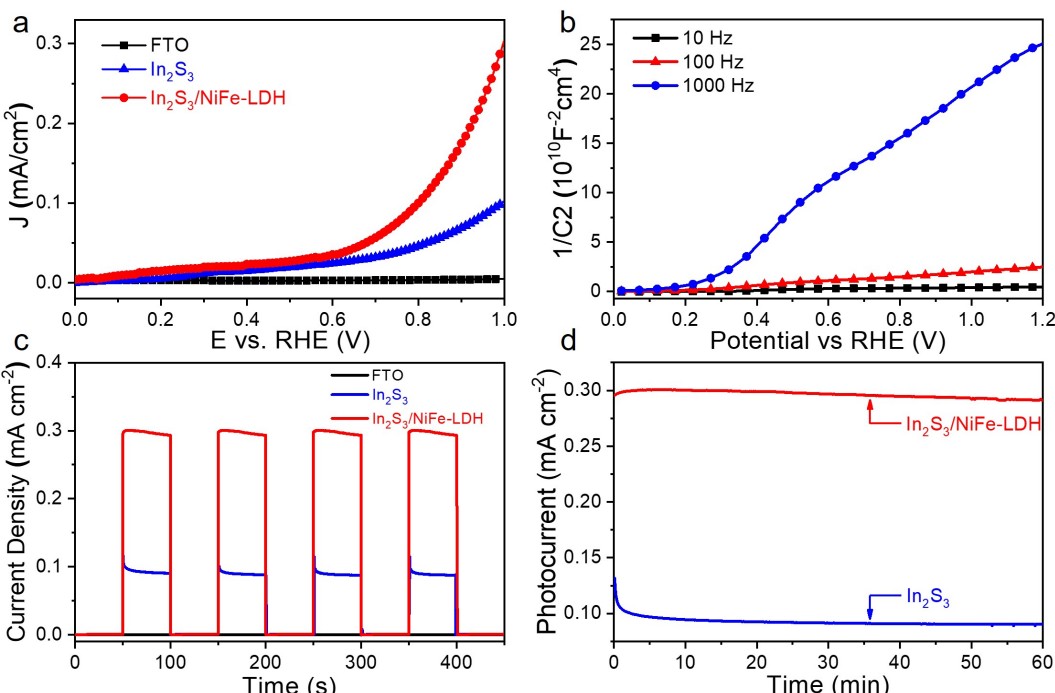

**Figure 5.** Effect of NiFe-LDH on PEC performance of the In$_2$S$_3$/NiFe-LDH photoanode. (**a**) The LSV curves, (**b**) Mott−Schottky plots of In$_2$S$_3$/NiFe-LDH, (**c**) I−t curves at a potential of 1.0 V vs. RHE under AM 1.5 global irradiation in a 0.25 M Na$_2$SO$_4$ solution, (**d**) time current stability a potential of 1.0 V vs. RHE under AM 1.5 global irradiation in a 0.25 M Na$_2$SO$_4$ solution.

The transient current responses of In$_2$S$_3$ and In$_2$S$_3$/NiFe-LDH photoelectric materials were measured using the chronocurrent method shown in Figure 5c. AM 1.5 G simulated sunlight was used to illuminate optoelectronic material, and the light source was switched every 50 s. When the light source is switched on, the transient current of In$_2$S$_3$/NiFe-LDH optoelectronic material can reach 0.3 mA/cm$^2$, while that of In$_2$S$_3$ optoelectronic material can only reach 0.1 mA/cm$^2$ in four cycles. Another important performance of PEC battery applications is photochemical stability. Figure 5d shows the photocurrent stability (relative to the time curve) of In$_2$S$_3$ and In$_2$S$_3$/NiFe-LDH optoelectronic materials under AM 1.5 G simulated sunlight. In Figure 5d, In$_2$S$_3$ and In$_2$S$_3$/NiFe-LDH photoelectricity materials still kept good stability with different photocurrent after 1 h irradiation, indicating that the In$_2$S$_3$/NiFe-LDH photoelectricity materials not only have high photoelectricity but also have good photostability and chemical stability in electrolyte solution (SO$_3^{2-}$).

In order to further investigate the influence of heterojunction on the charge separation process and electronic properties, the electrochemical impedance spectroscopy (EIS) of In$_2$S$_3$ and In$_2$S$_3$/NiFe-LDH photoelectrodes was measured in Figure 6. The Nyquist diagram obtained by In$_2$S$_3$ and In$_2$S$_3$/NiFe-LDH at open circuit potential is shown in Figure 6. The proposed Randles circuit of both In$_2$S$_3$ and In$_2$S$_3$/NiFe-LDH is in good agreement with the experimental data points (fitting error ~0.01). In addition, the EIS curve of In$_2$S$_3$/NiFe-LDH photoanode shows a smaller diameter than that of In$_2$S$_3$ photoanode under illumination, meaning that the resistance of In$_2$S$_3$/NiFe-LDH electrolyte interface is much smaller than that of the In$_2$S$_3$ electrolyte interface [60], which demonstrates that the NiFe-LDH nanosheets can effectively promote the charge transfer between the photoelectrode/electrolyte interface and thus enhance the photoelectric properties of In$_2$S$_3$/NiFe-LDH materials.

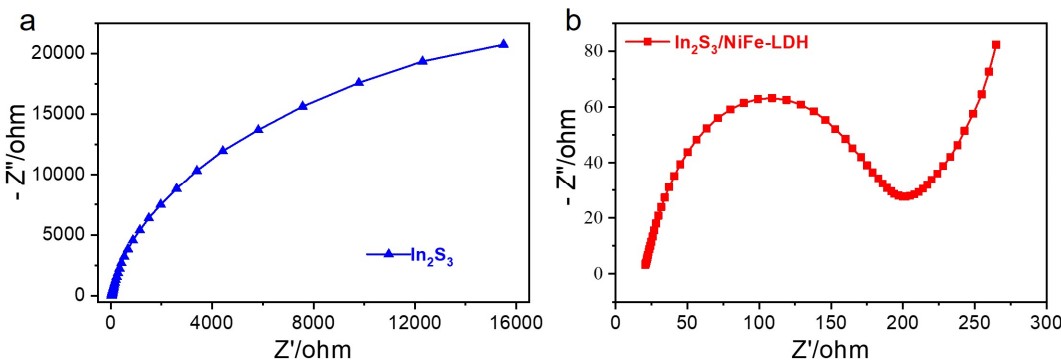

**Figure 6.** EIS spectra of $In_2S_3$ (**a**) and $In_2S_3$/NiFe-LDH (**b**) measured at the open−circuit potential under illumination.

### 2.6. Photodegradation of Xylose

The redox ability of the $In_2S_3$/NiFe-LDH material was obtained by the oxidation and degradation of the xylose standard solution. At room temperature, a three-electrode system was used, and the light source was AM 1.5 global irradiation. A 30 mL xylose standard solution and a 30 mL sodium sulfate solution were added to the electrolytic cell, and the xylose standard solution was oxidatively degraded under constant stirring at a constant battery voltage of 0.2 V vs. RHE. During the progress, the hydrogen evolution reaction ($2 H_2O + 2 e^- \rightarrow H_2 + 2 OH^-$) occurred at the cathode, and the oxidation of xylose to xylic acid ($C_5H_{10}O_5 + 3 OH^- \rightarrow C_5H_9O_6^- + 2 H_2O + 2 e^-$) occurred at the anode. The possible photo-induced electron–hole separation processes of samples are shown in Figure 7. The photogenerated electrons on the surface of samples transfer into the conduction band under light illumination. The electrons move quickly to the cathode, and a hydrogen evolution reaction occurs under applied voltage; meanwhile, the xylose oxidizes to xylic acid at the anode. Thus, the photogenerated electrons and holes could be separated effectively. In this way, the recombination of electron–hole pairs can be reduced, resulting in excellent photoelectrocatalytic activity. The residual xylose in electrolytic cells was determined using the phloroglucinol method: the aldehyde group in the lignin reacts with phloroglucinol to form a purple-red substance, which has an absorption peak at 554 nm [61]. Xylose solutions were sampled near the $In_2S_3$/NiFe-LDH electrode material at given irradiation time intervals of 30 min. After 3 h, the xylose content near the $In_2S_3$/NiFe-LDH electrode material was significantly reduced.

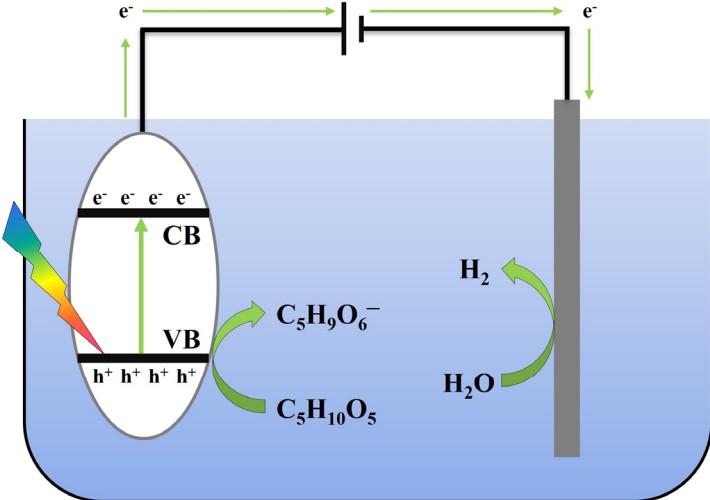

**Figure 7.** Schematic description of the photoelectrocatalytic mechanism of samples.

The amount of xylose remaining was also measured in the solution every 30 min. Figure 8a reveals the degradation of xylose in the photodegradation solution under UV, Vis, and NIR lights, and the photo-oxidation effect under UV light is the best. The xylose near the In$_2$S$_3$/NiFe-LDH electrode material is almost completely degraded, with a degradation rate of over 90% under UV light after 3 h, while the degradation effect of xylose under visible light applied voltage reached about 80%. In addition, the effect of simple voltage on the degradation of xylose was also explored. It is obvious that the simple electrocatalytic degradation of xylose was not very satisfactory, and the degradation rate was about 10% after 3 h. The oxidative degradation of the xylose solution by photoelectric materials under pure illumination was also carried out, as shown in Figure 8b, and the results indicated that the degradation effect of xylose by simple illumination is directly proportional to the effect of photocatalytic degradation of xylose. The effect of applied voltage on the photodegradation of xylose was determined in combination with Figure 8a,b; it can be concluded that the applied voltage can promote the transfer of photogenerated carriers, accelerate the degradation effect, and facilitate the reaction more easily. Figure 8c displays the temporal evolution of the absorbance changes at a time interval of 30 min. During the reaction of photoelectrocatalytic degradation, the residual xylose content was gradually reduced to the wavelength of 554 nm and kept stable ultimately after 3 h.

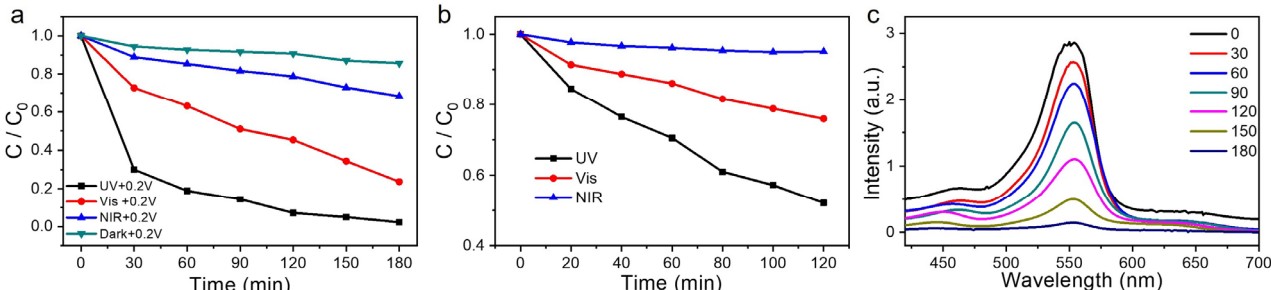

**Figure 8.** Photodegradation of xylose under (**a**) UV, visible, and NIR lights at 0.2 V and (**b**) UV, visible, and NIR lights, respectively. (**c**) The temporal evolution of the spectra during photoelectrocatalytic degradation of xylose with In$_2$S$_3$/NiFe-LDH under UV light.

Finally, the products after the degradation reaction of xylose were also examined. Firstly, the degradation products were purified by distillation and concentrated and freeze-dried. The analysis by [1]H NMR of the degradation reaction products was used to confirm the nature of the compounds formed at the anode. Figure 9 provides the [1]H NMR spectra for the initial xylose solution before (black curve) and after (red curve) photodegradation normalized with respect to the internal reference, in which the [1]H NMR spectrum of the initial xylose solution shows typical signals for xylose in water with several peaks in the 3.0 to 3.6 ppm range; the two doublets at $\delta = 4.4$ ppm and $\delta = 5.1$ ppm are the typical hydrogens on the anomeric carbon of the $\alpha$ and $\beta$ forms, respectively, and a quadruplet of peaks at $\delta = 3.75$ ppm are attributed to the hydrogen on the carbon next to the anomeric carbon of the $\beta$ form of xylose [43]. After photodegradation for 3 h, the xylose signal was lowered, and the total amount and position of the peaks in the range of 3.0 to 3.8 ppm changed. Typical peaks belonging to xylogous acid neared $\delta = 3.8$ and $\delta = 3.9$ ppm appeared.

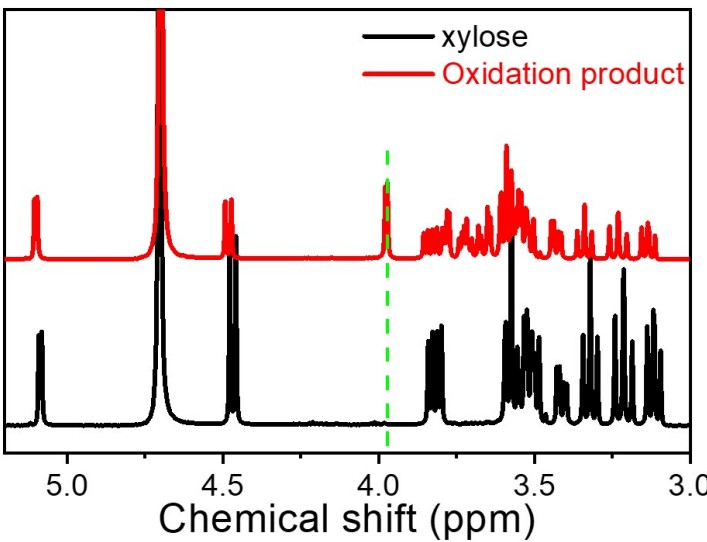

**Figure 9.** $D_2O$ $^1H$ NMR of xylose and oxidation product.

## 3. Materials and Methods

### 3.1. Materials

All chemicals used in the experiments were of analytical grade and were used as purchased without further purification. $InCl_3$ and $C_5H_{12}O_5$ were purchased from the McLean Reagent. Purified water (18.2 MX cm) for the experiments was attained from a Molecular Lab water purifier. The fluorine-doped tin oxide-coated glass (FTO, Zhuhai Kaivo Optoelectronic Technology Co., Ltd. (Zhuhai, China), <15 X/sq) was first cleaned with a detergent, boiled in a mixed solution of 30% ammonia water, 30% hydrogen peroxide, and ultrapure water for 30 min, then ultrasonically in ultrapure water and pentahydrate ethanol for 15 min, and finally sealed in anhydrous ethanol for later use.

### 3.2. Synthesis of $In_2S_3$ Electrodes Film

Firstly, the proper amount of citric acid coordination agent was dissolved in deionized water, and then 0.001 mol $InCl_3$ ($InCl_3$: citric acid = 1:6) was added into the solution continuously to obtain a completely clear and transparent solution, whose pH was about 2. Then, a $CH_3CSNH_2$ solution ($InCl_3$: $CH_3CSNH_2$ = 1:4) was slowly added to the above solution and continuously stirred to obtain a transparent precursor solution, whose apparent concentration of $In_2S_3$ was 0.01 mol/L. Then, the precursor solution was transferred into the reaction vessel, and the pre-treated F-doped $SnO_2$ glass substrate was also placed in the above solution. Finally, the reaction vessel was kept at 80 °C for 8 h, and then the resultants were purified, cleaned, and dried to obtain an orange indium sulfide film.

### 3.3. Coating of NiFe-LDH Film on $In_2S_3$ Electrodes

NiFe-LDH films were grown on $In_2S_3$ electrodes using the hydrothermal method. Firstly, $Ni(NO_3)_2 \cdot 6H_2O$ (2.4 mmol), $Fe(NO_3)_3 \cdot 9H_2O$ (0.8 mmol), and $CO(NH_2)_2$ were dispersed in 35 mL deionized water, then the above solution with 5 mmol $In_2S_3$-FTO was transferred into a 50 mL Teflon-lined stainless-steel autoclave, which was sealed and maintained at 120 °C for 12 h. The products were subsequently washed using deionized water/ethanol three times and dried at 80 °C for 2 h.

### 3.4. Characterization

Powder X-ray diffraction patterns of the samples were collected on a Bruker (Billerica, MA, USA) D8 Advance powder diffractometer operating at 40 kV, 40 mA for Cu Kα radiation (λ = 1.54 Å) using a Cu Kα source, with a scan step of 0.02 and a scan range between 5° and 80°. The surface morphologies of the samples were characterized using scanning electron microscopy (SEM, Hitachi Regulus 8220, Tokyo, Japan) and transmission

electron microscopy (TEM, TF20, Electronics Co., Ltd. Jeol 2100F, Tokyo, Japan). X-ray photoelectron spectra (XPS) were measured using Thermo ESCALAB 250XI (Waltham, MA, USA) with X-ray monochromatization. The C1s peak at 284.8 eV was used as a reference for calibrating the binding energies of all the XPS spectra. The UV–Vis absorption spectra of the synthesized samples were measured using a near-infrared UV–Vis spectrophotometer (Cary (Addison, IL, USA) 5000 UV–Vis–NIR).

### 3.5. Photoelectrochemical Measurements

All photoanodes were measured as working electrodes by exposing an area of 1 cm$^2$ on a CHI660A electrochemical workstation. The three-electrode configurations, using Pt foil as counter electrodes and the Ag/AgCl electrode (3.0 M KCl) as a reference electrode, were adopted for photoelectrochemical measurements in a 0.25 M $Na_2SO_4$ solution. Linear sweep voltammograms (LSV) at a scan rate of 10 mV/s and amperometric i-t curves under chopped light were carried out under AM 1.5 G (100 mW/cm$^2$) ($\lambda > 420$ nm) illumination. Electrochemical impedance spectra (EIS) of $In_2S_3$/NiFe-LDH were performed in the frequency range of 100 kHz to 0.1 Hz, with an amplitude of 10 mV. Mott–Schottky analysis was carried out in a DC potential range of 0–1.2 V vs. RHE, with an AC potential frequency of 1 kHz under dark conditions. All measured potentials were converted to V vs. RHE (ERHE = $E_{Ag/AgCl}$ + 0.208 + 0.059 pH).

### 3.6. Photoelectrocatalytic Oxidation of Xylose

The oxidation degradation of xylose was carried out using the synthesized $In_2S_3$/NiFe-LDH. A 40 mL xylose solution with a concentration of 1 mg/mL and a $Na_2SO_4$ solution were added into a quartz container. Pt was used as an opposite electrode, Ag/AgCl was used as the reference electrode, and $In_2S_3$/NiFe-LDH was used as the working electrode. The content of residual xylose in the container was measured every 30 min, and the concentration of residual xylose was measured at 554 nm using a UV–Vis spectrophotometer (Shimadzu, Kyoto, Japan). The xylose solution was distilled and condensed, and the product was detected.

### 4. Conclusions

In summary, the hierarchical $In_2S_3$/NiFe-layered double hydroxide ($In_2S_3$/NiFe-LDH) nanoarrays on an F-doped $SnO_2$ glass substrate via a facile two-step method were synthesized and investigated. After being decorated with the NiFe-LDH nanosheets, the two-component photoanode $In_2S_3$/NiFe-LDH exhibited significantly enhanced photoelectrochemical properties compared with the $In_2S_3$ single-component; due to that, the NiFe-LDH nanosheets depositing on the surface of the $In_2S_3$ nanocrystal can reduce the accumulation of photogenic holes, facilitate the separation of photogenerated charge carriers, and enhance the light response and absorption. Furthermore, the prepared $In_2S_3$/NiFe-LDH semiconductor material was used for the photoelectrodes to degrade xylose and displayed excellent degradation performance under both ultraviolet light and visible light, with an oxidation rate higher than 90% within 3 h under UV light, indicating that the prepared $In_2S_3$/NiFe-LDH semiconductor material can successfully oxidize xylose into xylose acid, as confirmed by [1]H NMR. This work provides a promising approach for designing new heterojunctions to apply to biomass degradation.

**Author Contributions:** Conceptualization, methodology, software, and validation, Z.L.; formal analysis and investigation, Z.L. and X.L.; resources, data curation, X.L. and W.L.; writing—original draft preparation, Z.L. and X.L.; writing—review and editing, X.L. and W.L.; visualization, Z.S. and G.L.; supervision, D.Y.; project administration, H.W. and W.L.; funding acquisition, X.L., H.W. and W.L. All authors have read and agreed to the published version of the manuscript.

**Funding:** This research was funded by the National Natural Science Foundation of China (No. 22006082 and 52172147), the Natural Science Foundation of Shandong Province (No. ZR2022MB095 and ZR2021MB035), Shandong Province Key Research and Development Program (No. 2021ZDSYS18),

the QUTJBZ Program (No. 2022JBZ01-05), and the Integration Pilot Program of Science, Education, and Industry, and the Talent Scientific Research Project of Qilu University of Technology (No. 2023PY036 and 2023RCKY181), Foundation of State Key Laboratory of Biobased Material and Green Papermaking, Qilu University of Technology (No. GZKF202214, ZZ20210111 and ZZ20210101), Foundation of National Forestry and Grassland Administration Key Laboratory of Plant Fiber Functional Materials (No. 2020KFJJ01).

**Data Availability Statement:** The data presented in this study are available on request from the corresponding author.

**Acknowledgments:** This work was financially supported by the Shandong Environmental Protection Development Group Co., Ltd.

**Conflicts of Interest:** The authors declare that this study received funding from Shandong Environmental Protection Development Group Co., Ltd. The funder was not involved in the study design, collection, analysis, interpretation of data, the writing of this article or the decision to submit it for publication.

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
