# Peer review of "Excellent Photoelectro-Catalytic Performance of In2S3/NiFe-LDH Prepared by a Two-Step Method"

_catalysts, doi:10.3390/catal14040230_

Round 1

Reviewer 1 Report

Comments and Suggestions for Authors

The manuscript “Photoelectrocatalysis of In2S3/NiFe-LDH on oxidation of xylose” by

Xiaona Liu et al. concerns two important problems: (1) to purify waist water from lignocellulosic plant material through it transforming to useful chemicals and (2) to use the photo catalytic approach on a base of solar light for this purpose. The authors proposed an electrochemical oxidation-like method for oxidation of “model compound” xylose into xylose acid with the use of In2S3/NiFe-LDH catalyst supported on F-doped SnO2, as the photo catalyst. Very complicate catalyst preparation procedure was successfully realized that was proved by a number of physicochemical techniques.

There are several comments:

1.      Line 82. Replace “metal pd and pt…” with “metals Pd and Pt”.

2.      Lines 165-166. O element detected in XPS spectra could be oxygen from Ni – and Fe-hydroxides but not a result of “the absorption of oxygen or carbon dioxide on the surface of the In2S3/NiFe-LDH electrodes because of their exposure to the atmosphere.” More over: O 1s peak looks too symmetrical to be resolved to components.

3.      Line 266. Was instead of were.

4.      Line 288. It is necessary to clarify the figure caption for Figure 7 (c) and to give better description of this Figure 7(c) in the text.

5.      Line 290. The figure caption to Figure 8. Introduce “1H NMR spectra of..” in the beginning of caption.

6.      Figure 2f.The HRTEM image quality does not allow the interplanar spacing of 0.257 nm to be clearly seen.

7.      Figure 2g (EDS). All the letters are very small, they are impossible to read!

8.      It is necessary to add DOI to the reference list for each article in accordance with the journal’s rules.

9.      Line 337. Replace “Joel” with “Jeol”.

Comments on the Quality of English Language

Minor editing of English language required

Author Response

We thank the editor and reviewers for carefully reviewing our manuscript “Excellent Photoelectro-catalytic Performance of In2S3/NiFe-LDH prepared by a two-step method” which the title has been redrafted according to the comments of reviewer 3. We have revised the manuscript carefully according to the reviewer’s comments. Below is a point-by-point response to the reviewers’ comments.

Reviewer 2 Report

Comments and Suggestions for Authors

1.      In the Introduction part authors reported the possible chemical and electrochemical routes of xylose oxidation. However, no data on the photoelectrochemical oxidation of xylose was given. It is fair to provide more details on the current research in the field of photoelectrochemical oxidation of organic substances including biomass-derived ones { For example, DOI: 10.1016/j.jallcom.2022.164276 }. After that, the novelty of the present paper should be highlighted.

2.      Page 5. For the determination of the band gaps of the synthesized materials the Tauc plot or Kubelka-Munk plot should be used.

3.      The PEC performance of NiFe-LDH nanosheet arrays loaded on the In2S3 photoanode should be shown both in Na2SO4 and Na2SO4+xylose.

4.      Please check the figure 5d.

5.      Page 8. Authors used the three-electrode cell for photodegradation of xylose. What does “at a constant battery voltage of 0.2 V vs. RHE” mean? Why the potential value of 0.2 V vs RHE was selected?

6.      The Figure 5 (a,c,d) is surprising. From Figure 5 a it is clear that the current value at 0.2 V vs RHE is very low. However, Figure 5 c, d demonstrates “I-t curves at a potential of 0.2 V vs. RHE under AM 1.5 Global irradiation in 0.25 M Na2SO4 solution (c)” where current values are much higher.

Comments on the Quality of English Language

Extensive editing of English language required

Author Response

(The authors gave the same response as above.)

Reviewer 3 Report

Comments and Suggestions for Authors

Review Report

Manuscript No.:      catalysts-2919534

Manuscript Title: Photoelectrocatalysis of In2S3/NiFe-LDH on oxidation of xylose

Authors: Xiaona Liu, ZhenZhen Li , Wenxia Liu , Huili Wang  , Zhaoping Song  , Dehai Yu , Guodong Li  

Authors report preparation synthesis of hierarchical In2S3/NiFe-layered double hydroxide (In2S3/NiFe-  LDH) nanoarrays on a F-doped SnO2 glass substrate via a two-step method which the In2S3  electrode film was firstly prepared using chemical bath deposition on F-doped SnO2 glass substrate, and then the layered NiFe-LDH was deposited on In2S3 electrode film by hydrothermal synthesis. They report a promising approach for designing new heterojunctions applying to biomass degradation. The manuscript can be accepted for publication after the Authors carry out Major revisions as appended below:

1.     Title should be redrafted

2.     Abstract to be modified with more significant results

3.     Experimental methods to be revised with necessary English corrections.

4.     Introduction, Page 3 , 108 to 104

Better these lines are redrafted which will indicate the proposed work and not significant results.

5.     Authors can give the reason for using NiFe-layered double hydroxide in the current work

6.      Scheme for oxidation of xylose into xylose acid in the current investigation to be provided

7.     2.2. SEM and TEM morphology characterization, Page 134 to 136

Authors state “The sample was rapidly dried in oven at 134 high temperature, resulting in slight damage and cracking of the NiFe-LDH on the surface 135 layer, and thus the double-layered material deposited on the FTO can be observed.”

Authors should state how cracking was confirmed.

8.     Line 146

“The EDS elemental analysis in figure 2g …..”

 Figure should be quoted in uniform pattern through  the article as per this journal pattern of presentation .

9.     Line 210-211, Page 6

As shown in 210 the figure. 5a,

The above line to be redrafted.

10.         Line 220

More details about Mott-Schottky curves to be included in the manuscript

11.         Line 240

The line “In figure 5d,……..” to be corrected

12.         Line  241

Authors state “In2S3 and In2S3/NiFe-LDH photoelectricity 240 materials kept good stability under 1 h irradiation

Why 1 hour irradiation was chosen by the authors?

13.         Line  249-246

figure 6.” to be corrected

14.         Whether authors used D-Xylose in this investigation?

15.         Concentration of Xylose and sodium sulfate solution to be provided

16.         Details of phloroglucinol method to be included in the experimental part of the manuscript.

17.         Line 272-274

Authors state “The xylose near the In2S3/NiFe-LDH electrode material is almost completely degraded with the degradation rate of over 90% after 3 h, and the degradation effect of xylose under visible light applied voltage reached about 80%.

Why xylose near the In2S3/NiFe-LDH electrode material is degrade by 90%  compared to 80% by visible light

18.         Line 277-279

The oxidative degradation of xylose solution by photoelectric  materials under pure illumination was also carried out to detect the effect of applied  voltage on photo-degradation of xylose as shown in figure 7b,”

These lines to be redrafted

19.         Figure 7

Figure quality to be improved.

20.         Whether authors confirm xylose and xylose acid by 13C NMR.

21.         3.2. Synthesis of In2S3 electrodes film, Line 319

Why the synthesis carried out at pH =2

22.         UV-Vis spectrophotometer to be included

23.         Conclusion to be redrafted

25..    Mechanism for oxidation of xylose should be included in the manuscript

25.Authors are informed to include some more recent references

Comments on the Quality of English Language

English language can be improved

Author Response

(The authors gave the same response as above.)

Round 2

Reviewer 2 Report

Comments and Suggestions for Authors

Some of the important comments were not taken into consideration in preparing the revised paper.

1.                  The optical diffuse reflectance data of a semiconductor material is usually converted into the Kubelka–Munk function before proceeding to process the conventional Tauc’s plot from which optical bandgap energy can be determined [1007/s12596-024-01741-0].

2.                  If authors use three-electrode cell for photoelectrodegradation of xylose (as authors claim in 3.6. “Pt was used as opposite electrode, Ag/AgCl was used as reference electrode, and In2S3/NiFe-LDH was used as working electrode”) the potential of the working electrode with respect to the potential of the reference electrode (in your case Ag/AgCl reference electrode) is controlled. Cell potential (voltage) is the charge difference between two electrodes (working and counter).

3.                  In order to select the oxidation potential of xylose for further photoelectrodegradation experiments authors should record the LSV curves for Na2SO4+xylose (under what potential xylose starts to oxidize under light). Photocurrent investigations should be performed to show the possible current multiplication effect in the present of xylose [10.1016/j.jelechem.2015.05.029]. Authors claim that “The sample exhibits photocatalytic activity in the absence of voltage, therefore, just adding 0.2 V voltage can promote the separation of photogenerated charge and improve performance”. However, the paper is devoted to Photoelectro-catalytic Performance of In2S3/NiFe-LDH2 so it should be proved by photoelectrochemical measurements.  Otherwise, the evidence on the excellent Photoelectro-catalytic Performance of In2S3/NiFe-LDH 2 is not convincing enough.

Comments on the Quality of English Language

Extensive editing of English language required.

Author Response

We thank the editor and reviewers for carefully reviewing our manuscript “Excellent Photoelectro-catalytic Performance of In2S3/NiFe-LDH prepared by a two-step method” which the title has been redrafted according to the comments of reviewer 3. We have revised the manuscript carefully according to the reviewer’s comments. Below is a point-by-point response to the reviewers’ comments.

Reviewer 2:

Some of the important comments were not taken into consideration in preparing the revised paper.

  1. The optical diffuse reflectance data of a semiconductor material is usually converted into the Kubelka–Munk function before proceeding to process the conventional Tauc’s plot from which optical bandgap energy can be determined [1007/s12596-024-01741-0].

Response: Thanks for your excellent suggestion. The optical diffuse reflectance data of a semiconductor material is usually converted into the Kubelka–Munk function, (αhv)1/n = A(hv - Eg), in which n depends on the type of semiconductor, the n of a direct bandgap semiconductor is 1/2, and that of an indirect bandgap semiconductor is 2。However, it is difficult to determine the type of composite materials, and then the value of n cannot be determined. Another way to calculate the band gap of materials is the transversal method. The basic principle of the transversal method is that the band edge wavelength (λ) of the semiconductor is determined by the band gap width Eg, and they follow the relationship Eg= 1240/λ, which we used in this work.

  1. If authors use three-electrode cell for photoelectrodegradation of xylose (as authors claim in 3.6. “Pt was used as opposite electrode, Ag/AgCl was used as reference electrode, and In2S3/NiFe-LDH was used as working electrode”) the potential of the working electrode with respect to the potential of the reference electrode (in your case Ag/AgCl reference electrode) is controlled. Cell potential (voltage) is the charge difference between two electrodes (working and counter).

Response: Thanks for your excellent suggestion. We failed to understand the reviewer’ meant or is there any question need to be answer here?

  1. In order to select the oxidation potential of xylose for further photoelectrodegradation experiments authors should record the LSV curves for Na2SO4+xylose (under what potential xylose starts to oxidize under light). Photocurrent investigations should be performed to show the possible current multiplication effect in the present of xylose [10.1016/j.jelechem.2015.05.029]. Authors claim that “The sample exhibits photocatalytic activity in the absence of voltage, therefore, just adding 0.2 V voltage can promote the separation of photogenerated charge and improve performance”. However, the paper is devoted to Photoelectro-catalytic Performance of In2S3/NiFe-LDH so it should be proved by photoelectrochemical measurements. Otherwise, the evidence on the excellent Photoelectro-catalytic Performance of In2S3/NiFe-LDH is not convincing enough.

Response: Thanks for your excellent suggestion. The evidence on the excellent Photoelectro-catalytic Performance of In2S3/NiFe-LDH were provided in figure 7. The sample exhibited photocatalytic activity in the absence of voltage, and applying a voltage of 0.2V can promote the separation of photogenerated charge and improve performance, which is enough to prove the sample has excellent Photoelectro-catalytic Performance. Even a higher voltage is applied, better performance may achieve, while the cost will also increase. By applying low voltage to improve the oxidation efficiency and reduce the cost is our ultimate goal.

Reviewer 3 Report

Comments and Suggestions for Authors

Authors report preparation synthesis of hierarchical In2S3/NiFe-layered double hydroxide (In2S3/NiFe-LDH) nanoarrays on a F-doped SnO2 glass substrate via a two-step method which the In2S3 electrode film was firstly prepared using chemical bath deposition on F-doped SnO2glass substrate, and then the layered NiFe-LDH was deposited on In2S3 electrode film by hydrothermal synthesis. They report a promising approach for designing new heterojunctions applying to biomass degradation. Authors have carried out extensive revisions in the manuscript and  responded to the queries raised. Now the manuscript can be published in this esteemed  journal.

Author Response

Thank you very much for your suggestions and decision.